# The Potential of AR Solutions for Behavioral Learning: A Scoping Review

**Crispino Tosto** [1,*], **Farzin Matin** [2], **Luciano Seta** [1], **Giuseppe Chiazzese** [1], **Antonella Chifari** [1], **Marco Arrigo** [1], **Davide Taibi** [1], **Mariella Farella** [1,3] **and Eleni Mangina** [2]

1   Istituto per le Tecnologie Didattiche, Consiglio Nazionale delle Ricerche, 90146 Palermo, Italy;
    luciano.seta@itd.cnr.it (L.S.); giuseppe.chiazzese@itd.cnr.it (G.C.); antonella.chifari@itd.cnr.it (A.C.);
    marco.arrigo@itd.cnr.it (M.A.); davide.taibi@itd.cnr.it (D.T.); mariella.farella@itd.cnr.it (M.F.)
2   School of Computer Science, University College Dublin, D04 V1W8 Dublin, Ireland;
    farzin.matin@ucdconnect.ie (F.M.); eleni.mangina@ucd.ie (E.M.)
3   Department of Computer Science, University of Palermo, 90128 Palermo, Italy
*   Correspondence: crispino.tosto@itd.cnr.it

**Abstract:** In recent years, educational researchers and practitioners have become increasingly interested in new technologies for teaching and learning, including augmented reality (AR). The literature has already highlighted the benefit of AR in enhancing learners' outcomes in natural sciences, with a limited number of studies exploring the support of AR in social sciences. Specifically, there have been a number of systematic and scoping reviews in the AR field, but no peer-reviewed review studies on the contribution of AR within interventions aimed at teaching or training behavioral skills have been published to date. In addition, most AR research focuses on technological or development issues. However, limited studies have explored how technology affects social experiences and, in particular, the impact of using AR on social behavior. To address these research gaps, a scoping review was conducted to identify and analyze studies on the use of AR within interventions to teach behavioral skills. These studies were conducted across several intervention settings. In addition to this research question, the review reports an investigation of the literature regarding the impact of AR technology on social behavior. The state of the art of AR solutions designed for interventions in behavioral teaching and learning is presented, with an emphasis on educational and clinical settings. Moreover, some relevant dimensions of the impact of AR on social behavior are discussed in more detail. Limitations of the reviewed AR solutions and implications for future research and development efforts are finally discussed.

**Keywords:** augmented reality; behavioral learning; social interaction; social behavior; scoping review

## 1. Introduction

In augmented reality (AR), digitally created content is superimposed over the user's real-world environment using a device (e.g., cellphone) that incorporates real-time inputs in an attempt to enhance the user's experience. According to Billinghurst [1] and Shelton [2], to create an AR experience, digital and virtual objects (e.g., graphics and sounds) are superimposed over an existing environment. In other words, AR applications merge virtual or computer-generated content with the real world [3]. Currently, AR is accessible from multiple devices: traditional computers, tablets, mobile phones and, increasingly, wearable devices such as AR headsets (HMDs) and smart glasses, which offer users a more realistic interaction with AR objects without the need to hold the device.

In the last decade, educational researchers and practitioners' interest in emerging technologies, such as AR, has increased significantly, and new opportunities for teaching and learning processes have been explored. In the educational field, studies indicate that AR solutions can ameliorate students' academic achievement compared with traditional teaching and learning methods [4]. The existing research also confirms that AR solutions

have a greater impact on learners' experiences in terms of content understanding and retention, interest, engagement, and satisfaction with the learning material than traditional and different digital media-related learning experiences [5,6]. Additionally, two recent reviews described advancements and benefits in the use of AR in primary and secondary education [7,8]. These studies provide relevant indications for the use of AR game-based learning to enhance students' positive attitudes toward learning, participation, knowledge transfer, and skill acquisition. The prevalent use of AR for educational purposes concerns the acquisition of content knowledge and cognitive skills related to the broad field of natural sciences among students of all grades. On the contrary, only a limited number of studies explored the support provided by AR to the learning of content related to social science disciplines such as psychology and health and welfare [4,5]. The literature has also described and assessed the use of technology (AR in particular) in clinical care settings [9] and its application to the treatment of psychological disorders [10,11]. With regard to professional settings, a relatively recent review [12] identified knowledge acquisition, behaviors and practical skills, and affective dimensions as the main outcomes AR-supported training focuses on. However, to the best of our knowledge, the literature has not directly addressed the state of the art of AR solutions within interventions specifically designed for teaching or training behavioral skills across different settings (e.g., educational, clinical, and professional). The identification of strengths on the one hand and research gaps and limitations on the other in this field may provide relevant information to researchers and practitioners interested in designing effective AR solutions for behavioral skills learning, regardless of the specific setting of intervention.

Furthermore, the literature has shown that individuals who use different types of AR applications are more open to expected behavioral changes [13]. Augmented reality's features allow for the creation of a potential platform for behavior change or the influence of social activities and routines. Research has offered important insights into the connection between AR and behavior change. Many studies concentrate on the technological and development features of AR, but limited studies have researched how technology affects social experiences, in particular the impact of using AR on social behavior and the effect of AR interventions on behavioral outcomes. Referring to a review paper by Kim et al. [14], only 9 out of 526 studies explored some aspect of social interaction, which is less than 2%.

With this perspective in mind, the purpose of this scoping review is to fill these research gaps by exploring research that investigates the impact of AR technology on behavioral change. First, this work aims to intercept studies that apply AR in promoting the teaching and learning of behavioral routines and skills, with a special look at educational and clinical settings. In addition, the study undertakes an exploration of the context of AR technology and its ability to influence social behavior.

## 2. Methodology

For the research purposes of this study, a scoping literature review has been conducted, given that the area of research is new and investigating the use of AR within interventions designed to teach and train behavioral skills and its impact on social behavior and interaction has not been studied in detail [15]. Unlike systematic reviews, scoping studies usually do not require a formal evaluation of the quality of the methods underlying the selected studies However, as this is a new field of research, scoping reviews are an excellent instrument to determine the scope of coverage of a range of information on a given subject and provide a clear indication of the amount of the available literature and studies, as well as an outline of its focus and the types of evidence and research gaps [16,17].

### 2.1. Research Questions

In order to clarify the research questions, we conducted a research strategy based on the Population, Concept, Context (PCC) framework. The objective of this review was to explore the use of state-of-the-art AR solutions in interventions to promote the acquisition of behavioral skills, irrespective of the specific population targeted and the

specific intervention setting. An additional review goal was to look at some of the key aspects of using AR technology and its effect on social behavior and social interactions.

Accordingly, this study sought to answer the following research questions:

RQ.1 How are AR systems designed in the context of interventions (i.e., educational, clinical, and professional) for teaching or training a behavior?

RQ.2 What is the impact of using AR on social behavior?

*2.2. Scoping Review Procedure*

The first step of the review included identification of the search strings for the selection of relevant papers. The search strings were edited based on the two main research questions mentioned above (RQ.1 and RQ.2).

In order to be included in the review, the papers needed to meet the following inclusion criteria:

- Papers from any country (written in English);
- Papers published since 2010;
- Full text available;
- Given the paucity of experimental or quasi-experimental studies on the review topics, general discussion, and theoretical papers, case studies, examples of applications, reports, and conference proceedings were included;
- Peer-reviewed papers.

Additionally, papers that mentioning AR but referring to mixed reality (MR) or virtual reality (VR) were excluded. The papers were then screened based on their titles and abstracts to remove those not relevant to the research questions. For the selection of articles, the PRISMA protocol was adapted for the purpose of the scoping review [18,19], and the results will be presented later in this paper.

2.2.1. How Are AR Systems Designed in the Context of Interventions for Teaching or Training a Behavior?

With regard to the first question, more specific research questions were posed to guide the review process:

- Which are the examples of the use of AR within interventions focused on behavioral teaching or training?
- What are the most common settings for these interventions?

In order to answer these questions, the following search string was implemented: (("augmented reality") AND (behav*) AND ((education*) OR (train*) OR (teach*) OR (learn*))). The string was used and adapted to retrieve papers in the following three databases: PubMed, Scopus, and Web of Science. Additionally, papers were found in the OpenAIRE database using the following search string: "augmented reality AND behavior".

For the selection of papers related to this research question, two further exclusion criteria were added to the criteria previously listed: (1) papers exploring the acceptance of technology and user behavior when dealing with technology were excluded, and (2) studies with clinical patients not concerning the training of socially relevant behavior were excluded. Figure 1 shows the PRISMA protocol for the selection of papers relevant to the first research question of the scoping review. The figure also includes the used search string.

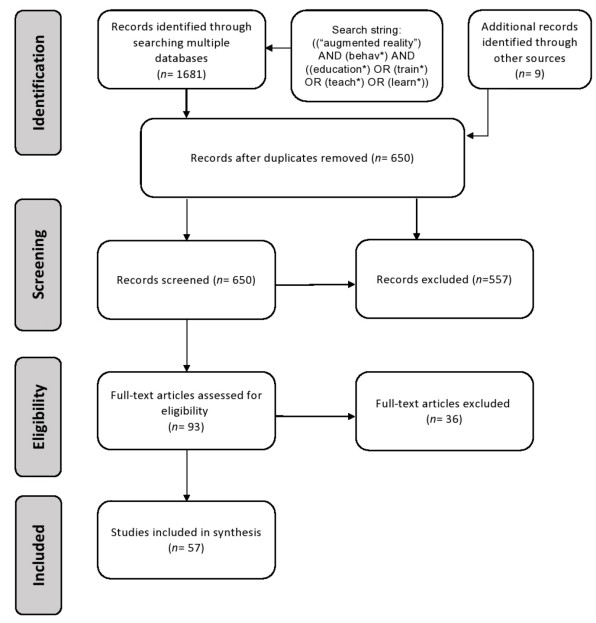

The * symbol represents the truncation symbol commonly used in most scientific databases to include various word endings and spellings.

**Figure 1.** PRISMA protocol for articles relevant to the first research question.

Once the study selection was completed, salient information was extracted from the papers selected to answer the first research question with respect to the following dimensions:

- Title of publication;
- Year of publication;
- Type of publication;
- Keywords;
- Setting of intervention (educational, clinical, or professional);
- Peer-reviewed papers;
- Target population:
  - Age and grade for academic settings;
  - Age and disease for clinical settings;
- Type of AR technology;
- Purpose of AR application;
- Theoretical framework of the behavioral intervention;
- Key findings.

2.2.2. What Is the Impact of Using AR on Social Behavior?

For the second research question, the following string was used: (("augmented reality") OR (ar) OR (("smart glasses") OR ("google glass") OR ("smartglasses"))) AND (socia*) AND (behav*). This was used for identifying relevant papers in five electronic databases: PubMed, ACM Digital Library, Web of Science, IEEE Xplore, and Scopus.

Figure 2 summarizes the steps followed for the papers identified and selected for the second research question. The search string for identification of relevant papers has been included as well.

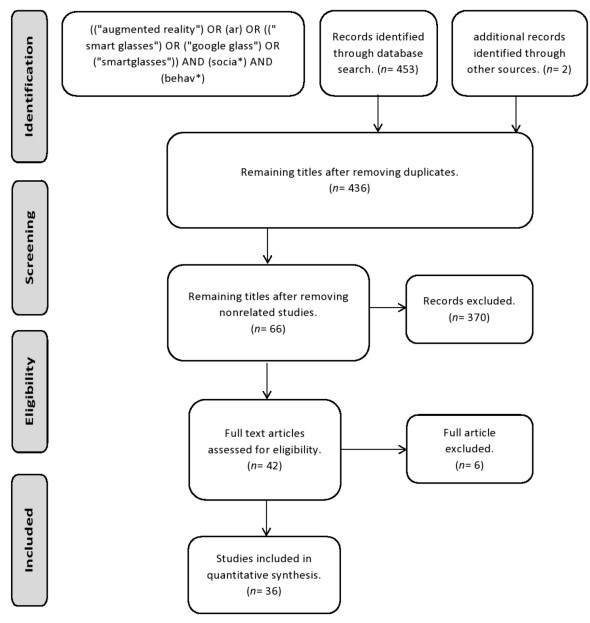

The * symbol represents the truncation symbol commonly used in most scientific databases
to include various word endings and spellings.

**Figure 2.** PRISMA protocol for articles relevant to the second research question.

To address the second research question, the features and information of the selected papers were extracted into a table to provide a detailed overview of the documents included. The following is a list of the critical information charted for each paper:

- Title of publication;
- Year of publication;
- Study purpose;
- Participants (age group and number of participants);
- Technology used (Software and hardware);
- Assessment method;
- Key findings.

## 3. Results

As shown in Figures 1 and 2, a total of 93 articles were included in the current paper. A total of seven articles selected for the second research question were found to be overlapping with those identified to answer the first research question. As a consequence, it should be considered that a total of 86 papers were actually reviewed and charted for the purpose of the current work. The following sections summarize the results of the current study based on the questions that guided the research.

### 3.1. The Use of AR within Interventions Designed for Teaching or Training a Behavior

In this section, we describe the papers selected to answer the first research question of the scoping review. First of all, we present the distribution of papers related to the use of AR within interventions designed to promote the acquisition of behavioral skills per setting of intervention (Table 1).

**Table 1.** Number of articles selected per setting of intervention.

| Setting | N | Reference |
|---|---|---|
| Educational | 26 | [20–45] |
| Clinical | 17 | [46–53,53–62] |
| Professional | 5 | [63–67] |
| Other | 9 | [68–76] |

N = number of selected articles.

Most of the papers describe studies related to the impact of AR technology on social behavior conducted in educational settings. These studies are followed by articles presenting AR solutions designed for clinical settings. It is noteworthy that many studies, even if conducted in the school context, focused on subjects with specific behavioral difficulties, mainly autistic spectrum disorder (ASD). Finally, the scientific production investigating the impact of AR technology on behavior training and management in work environments is marginal and limited to the last few years.

Figure 3 shows the evolution over time of the number of papers differentiated per setting of intervention.

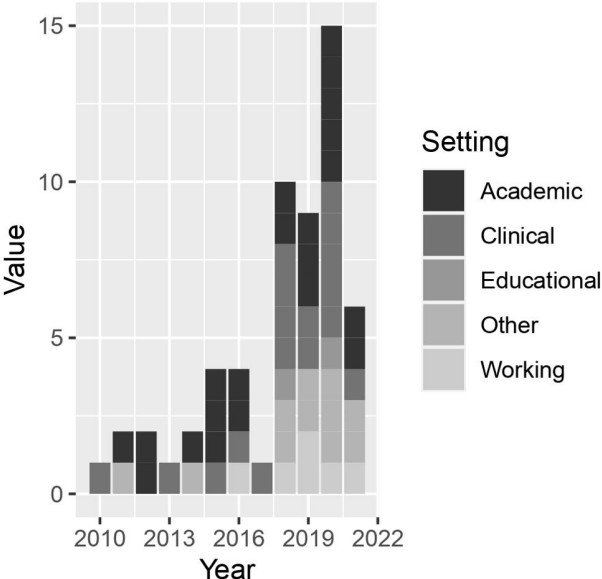

**Figure 3.** Number of papers per year and setting of intervention.

Looking at the trend of the selected publications, a marked increase in the number of published papers from 2016 can be observed. The year 2020 demonstrates a relevant number of published papers in both educational and clinical settings.

3.1.1. Interventions Delivered in Educational Settings

The distribution of the studies conducted in educational settings and differentiated per school grade is summarized in Table 2. Studies involving primary school students prevailed, along with those targeting secondary school students. (Some articles covered multiple studies at different school grades or descriptions of AR solutions for students of different grades.) According to these studies, AR technology was mainly used to support the learning process of basic social and relational skills. It also had a significant number of applications for students diagnosed with ASD [23,27,31,33,37,39,45].

**Table 2.** Number of selected studies per school grade.

| Setting | N | Reference |
| --- | --- | --- |
| Early | 2 | [28,35] |
| Primary | 13 | [20,25–27,29,31,33,34,37–40,43] |
| Lower secondary | 7 | [20,22–24,34,37,40] |
| Upper secondary | 4 | [20,37,40,41] |
| Tertiary | 6 | [21,25,30,32,42,44] |
| Not specified | 2 | [21,25,30,32,42,44] |

N = number of selected articles. Some articles covered multiple studies conducted with samples of different school grades or presented educational solutions for students of different grades.

With regard to the most used devices for the utilization of AR content in the scholastic and academic settings, smartphones and tablets prevailed (17 papers out of the 26 selected), and only in rare cases were PCs and webcams still used [25,33,41]. Few works introduced wearable devices such as smart glasses [23], HoloLens [43], and helmets [32] or integrated advanced technologies such as the kinectical skeletal tracking system to AR solutions [26]. Almost all of the reviewed papers described marker-based AR solutions using both paper markers and objects as triggers [27,28,31,36]. Only a few papers [25,26,39,43,77] presented more complex solutions based on dynamic interactive augmentation [78], mixing object recognition and motion tracking for a location-based solution [24]. This is probably due to the fact that the introduction of advanced devices and technologies is not always feasible in the educational field. This is characterized by a large number of stakeholders, as well as a lack of financial resources and specific expertise, both of which are necessary to handle innovative technologies.

AR solutions were mainly used as tools to support the learning and training of prosocial [28,29,31,33,39,44], empathetic [43], and anti-corruption [21] behaviors for environmental education [20,24,38,41] and for the training of specific motor skills [25].

Concerning the theoretical framework, some of these papers were based on traditional approaches to the treatment of behavior such as behavioral change [20,23,24,26], social modeling [31,34], and cognitive behavioral [32,39] approaches. Other papers addressed behavioral training from the perspective of concept mapping [33], visual novel [21], theatre-based training [31], and coaching [37] or were inspired by more general theories, such as neuroconstructivism [28], the persuasion theory [24], and the theory of reasoned action [34].

In conclusion, the current review indicates that AR has been used within a plurality of methodological frameworks. This clearly demonstrates the flexibility of this technology. However, it also points to an evolving landscape in which the strengths and weaknesses as well as the opportunities and challenges related to its application in the educational setting are not always fully defined.

Surprisingly, only six of the reviewed papers directly addressed potential limitations or challenges related to the integration of AR technology in educational practice. Specifically, two papers [23,29] found that teachers and educators were reluctant toward modifications of their consolidated practices, especially due to the introduction of novel technologies such as AR. A further limitation on the use of AR in the educational field that emerged from the available research pertained to the availability of devices in terms of costs, level of implementation, and market readiness [23,29]. In this regard, two papers reported the design and technical limitations of the studied AR solutions, which were defined as context-unaware [39] and difficult to handle [43]. Gil et al. [43] suggested a minimization of the mechanical elements of AR systems to reduce the psychological pressure related to learners' worries about handling the required devices. An additional paper [26] noticed that setting characteristics and users' skills or aptitude may also influence the way an AR solution is actually used, limiting its potential for application. Finally, it should be noted that the novelty introduced by AR solutions can be so attractive that its actual educational potential is ignored or overlooked [29].

### 3.1.2. Interventions Delivered in Clinical Settings

Concerning papers reporting on interventions delivered within a clinical setting (Table 1), autism spectrum disorders (ASDs) were the most commonly treated or discussed disorders [46,47,51,53,55,59,60], followed by anxiety disorders (e.g., social anxiety disorder, generalized anxiety disorder, or phobias) [49,56,61] and attention-deficit hyperactivity disorder [53,54]. Additionally, a paper included in this category described a socioemotional learning intervention aimed at promoting the development of emotional coping and interpersonal skills among adolescents in a community setting [50].

Consequently, given the nature of the disorders considered, the papers mostly reported findings about AR solutions embedded into interventions designed for children [46,47,49,51–53,55,56,59,60], while young adults [53] and adults [48,58,61] were shown to be underrepresented.

When treating patients with ASD and ADHD, AR was used to enhance interventions designed to train behavioral and social skills (e.g., social interaction skills and emotion regulation), manage problem behaviors, and promote positive behaviors [46,47,53,55,59,60]. Concerning anxiety disorders, AR solutions were mainly implemented for the treatment of phobias and social anxiety through exposure practice [49,56,61]. The examined AR interventions mostly followed a cognitive behavioral approach [49,54,60,61] and used a video modeling strategy within the social learning framework [47,48]. Cognitive training [53] and a sequence learning framework [52] were also reported as theoretical and methodological frameworks underpinning the implemented interventions. Finally, AR solutions were developed for use with both portable (smartphone and tablet) and wearable devices (HoloLens and smart glasses) [51,52,59,60]. As for AR solutions designed for educational settings, most of the studies reported marker-based solutions, including markers, scenes, and object recognition [47,48,53,55,61,62]. An additional study combined marker-based technology with an example of non-specific digital augmentation [46], in which a digitalized dynamic view of the environment is provided but without a direct reference to what is currently being viewed [78]. Two papers described a location-based AR solution [50], with one of them reporting a combination with dynamic augmentation.

A number of limitations related to the use of AR should be mentioned. First of all, embedding AR technology into clinical interventions may be very expensive, especially if costs are covered by the patients [57]. Moreover, the development of effective solutions can require the use and integration of experimental technologies, which may cause system unreliability and cause problems for clinical patients, especially those diagnosed with anxiety disorders [56]. An additional paper highlighted the need for taking into careful consideration scalability issues when designing a new AR solution [62]. AR solutions may also be at risk of causing fatigue, headache, nausea, and eye strain, which are recognized as typical negative physical consequences associated with the use of AR technology [52]. Common concerns also relate to the risk of reduced attention, especially in children with specific diagnoses such as autism spectrum disorders or attention-deficit hyperactivity disorder [53]. It should also be noted that in the case of use with specific categories of clinical patients (e.g., patients with intellectual disabilities), environmental modifications or accommodations should be taken into account in order to facilitate the use of the developed AR solution [48]. Finally, some user interfaces might not be suitable for use in different cultural contexts [50].

### 3.1.3. Interventions Delivered in Professional Settings

With regard to the implementation of AR solutions for professional training (Table 1), it is noticeable that the selected papers were all published between 2018 and 2021. Thus, it seems that the use of AR technology for behavioral training is a relatively new research topic in this area. Applications ranged from medical applications [65,67] to the foodservice [63], green driving [64], and industrial assembly [66] fields. In this context, the use of innovative technological systems and devices was more frequent, as in the case of complex gaming systems that used head-up devices (HUDs) [64] or were equipped with sensors and kinetic

simulators [66]. The use of smart glasses [49] and head-mounted devices (HMDs) [66] was also frequent.

From a theoretical point of view, an approach that focuses on procedural training through segmentation into micro-tasks, as described by Lampen et al. [66], is common. On the other hand, a more general theoretical framework was presented in the work of Clark et al. [63], where reference was made to embodied cognition or grounded cognition, as well as in the work of Dixit and Sinha [65]. In these papers, the authors were interested in investigating the effectiveness of AR as a tool to facilitate the transfer of skills taught in specific behavioral training programs.

In summary, AR technologies were used in this context to simulate complex environments and define well-structured training programs with measurable performances, such as the assembly of car doors [66], handwashing procedures in a food service [49], or driving performance in terms of fuel consumption and pollution [64]. This is in contrast to research conducted in the educational field. In this research, AR solutions were typically designed and analyzed as tools to support learning by fostering students' attention, motivation, and retention.

### 3.2. The Impact of AR on Social Behavior

The results based on the analysis of the papers selected to answer the second research question indicate that although public interest in AR is new and increasing, academic researchers have been developing and researching the impact of this technology on human social behavior for decades. Miller et al. [72] studied social interaction in AR by designing three different experiments to examine the sociopsychological effects of AR. A well-known psychological hypothesis (i.e., social facilitation and inhibition) was introduced to an AR user with a virtual agent in the first experiment. The second experiment investigated whether users respond to social norms while dealing with virtual persons. It also investigated whether the spatial connections between physical locations and virtual information affected subsequent behavior. Lastly, the third experiment looked at the social costs of wearing an AR headset in comparison to people that are not wearing one. The participants interacted in dyads, and those who used AR headsets showed less emotional attachment to their partners than those who did not use AR headsets. To summarize their findings, the presence or absence of virtual content was shown to have a significant impact on task performance, nonverbal behavior, and social connectedness. Similarly, children's behavior can be significantly influenced by the visual input provided by AR technology [70]. In this study, before completing a filler task, two children aged 5–10 years old were shown a human-like AR character standing on one of the two physical routes. Following the task, the kids were asked to walk along one of two routes in order to receive a reward. Both children preferred the non-AR character pathway to the AR character pathway.

AR technology can also help young children's empathetic behavior, as it stimulates children's imagination and creativity without causing them to lose touch with reality. Empathy is one of the most critical factors in a child's ability to make friends at school and expand their social relationships. Gil [43] developed an AR storybook based on role-playing that allows children to learn empathy skills through an interactive reading environment in which they think and communicate in the voices of the story's characters.

### 3.2.1. Gamification in AR

Digital games with AR features have quickly become one of the most common types of entertainment in the world [79]. People who play digital games are more open to future improvements in their behavior [80]. AR technologies such as AR games can improve users' social interaction and behavior. Researchers observed a famous AR game, Pokémon GO, and discovered that AR games can have beneficial behavioral effects, including social interaction [81]. The entertainment interest in AR games is an influential aspect, and it becomes much more important when investigating AR game outcomes in the context of leisure time. Gamification in general has incorporated studies on technology and game de-

sign, inspiration, and human–computer interaction, among other things [82]. Arjoranta [83] highlighted another example of research on the different forms of behavior changes and their underlying game characteristics in the form of the popular AR game Pokémon GO. The study data were gathered using a survey of 262 Pokémon GO participants. The results showed that the questioned players adjusted their behaviors before or after playing Pokémon GO. The participants indicated that they were more social, expressed more positive feelings, found more value in their daily lives, and were more motivated to discover their environments.

Learners' behavioral intentions can be influenced by their perceptions of the AR learning system's efficacy and satisfaction. Chang's study [84] demonstrated how satisfied learners were with the AR-learning system, as well as their behavioral intentions to use the system and how effective it was.

According to the work performed by Kim [14], AR has the potential to overcome the lack of ability of intelligent virtual agents to provide nonverbal cues, which are an essential part of social interaction. This study's findings suggest that augmenting an agent with a visual body in AR and normal social behaviors could enhance the user's confidence in the agent's ability to affect the real world. Although many AR apps show embodied agents in scenes, no research into the social impact of these AR renderings has been undertaken. Jun [85] attempted to fill this research gap by investigating the social impact of simulated humans through two lenses: behavioral and anthropomorphic realism.

### 3.2.2. The Impact of AR on Social Behavior in Special Education

Children with special needs, especially those with autism spectrum disorders (ASD) [86], social communication disorders [87], and attention-deficit hyperactivity disorder (ADHD) [88] struggle to use appropriate communication techniques and skills in social interactions with their peers. We present here some articles describing the use of AR for clinical purposes in children with ASD and ADHD. Some of these articles were already summarized in the previous section (see Section 3.1.2), and they are discussed here in more detail. Moreover, the research conducted to answer the second question of this review made it possible to identify new papers in addition to those already found, some of which are introduced here as being of particular interest.

Based on Vahabzadeh's [89] work, such individuals can benefit from assistive AR smart glass technology. Their findings show that AR smart glasses can help students with ASD improve their feelings of anxiety, hyperactivity, and social withdrawal in a public elementary school environment. Furthermore, AR smart glasses could be an effective tool for meeting the behavioral needs of children diagnosed with ASD [77]. A study by Liu [90] showed that a specialized AR smart glass solution is practical, functional, and acceptable. Children diagnosed with ADHD, in comparison with their non-ADHD peers, have poor school and academic performance. By using AR smart glasses to support and enhance their abilities, ADHD-related symptoms in school-aged girls, teenagers, and young adults with ASD can be minimized, such as hyperactivity, inattention, and impulsivity [53]. One of the activities that has been proven to be especially beneficial in the treatment of children with social communication disorders is storytelling. Storytelling is crucial for the linguistic and cognitive growth of infants. The Chen AR model could help educate children on how to recognize and understand the emotions expressed in facial expressions. This could be performed in daily social interactions with children of all ages [47]. They can learn about body language and facial expressions through role-play. This teaching method can effectively improve the interactive social skills of children diagnosed with ASD. It can also reduce the fear and anxiety that they typically experience when they face real people. Role-playing will teach them about body language and facial expressions. This teaching approach will help children with specific needs develop their interactive communication skills while also reducing their fear and anxiety when they interact with regular humans [26].

## 4. Discussion

In recent years, AR has become increasingly popular as a useful piece of technology to deal with human behavior. This is both in terms of physiological aspects and in relation to specific problems and disorders. It should also be emphasized that the concept of AR itself is evolving as a result of the development of new devices and sophisticated integration with toolkits [91,92]. This scoping review was motivated by the increasing interest in this topic, and it was aimed at two specific research questions:

1. How do AR technologies impact the processes implied in learning, training, and modeling behavioral skills?
2. What is the impact of AR utilization on social behavior?

These two questions, although connected, focus on two slightly different issues related to AR popularity. In fact, they examine the impact of AR from two different perspectives: the first emphasizes the learning process, while the second highlights the social and psychological effects triggered by the interaction between users and technology. A total of 93 articles were selected to be included in the current study. However, 7 articles selected for the second research question overlapped those identified for the first research question [22,26,28,43,47,54,70]. Consequently, only 86 papers were actually reviewed for the purpose of the current study.

The application of AR technologies to learn and train behaviors and social skills is a relatively new topic. Even though the use of AR seems to positively influence students in learning actively, motivating them and thus leading to an effective process of learning (e.g., [4,5]), the application of these technologies to support behavioral learning programs is uncommon and often restricted to interventions for specific disorders such as ASD, ADHD, or social anxiety within educational settings (e.g., [31]). In clinical settings, it is possible to observe an increasing use of AR for designing interventions to train behavioral and social skills, manage problem behaviors, and promote positive behaviors or for the treatment of phobias and social anxiety. In working environments, AR gained increasing interest not only to train specific tasks and repetitive procedures but also as a tool to facilitate programs aimed at eliciting social behavior changes, also taking into consideration complex theoretical frameworks such as embodied cognition.

AR is an evolving bundle of hardware and software technologies that can be integrated with other innovations, such as gamification, sensors, artificial intelligence procedures, spatial mapping, and so on. For this reason, AR is flexible enough to support different phases and processes of behavioral learning and modeling. Moreover, the augmentation of reality permits dissemination in the environment of symbolic signs and marks able to guide and facilitate the process of the retention of procedures and tasks. In future research, the use of avatars can also support the learning of complex motor skills in a safe way, helping learners memorize procedures and reproduce routines. Additionally, the AR digital experience can be used to stimulate and enhance decision making and social problem solving skills in different social contexts. In summation, augmented reality can have an impact toward positive behavior not only within school and classroom settings but also within working and living environments. Research should investigate how we can define the world we live, study, and work in through the utilization of AR, and we can design these meaningful experiences with reusable requirements that can enrich humanity.

It is noteworthy that a relevant number of papers describing AR enhanced interventions for behavioral learning and training in both educational and clinical settings did not specify a theoretical framework guiding development efforts (e.g., [22,46]). Future research should be strongly encouraged to clearly define a rigorous framework upon which to design and validate AR solutions for behavioral training, especially in academic and clinical settings. Moreover, AR solutions designed for use at school were generally less advanced than those designed for clinical settings, with the latter including wearable devices such as HoloLens and smart glasses. On the one hand, this may clearly represent a limitation on the efficacy of interventions developed for educational purposes at school and should

be addressed by future research. On the other hand, it can be easily understood, given the high number of involved stakeholders in schools and lack of financial resources and the required expertise to handle more innovative technologies.

Interestingly, only a few papers explicitly discussed limitations pertaining to the integration of AR technology within behavioral interventions. Cost and availability, also in terms of the devices' technological and market readiness, were primary concerns of several papers describing AR solutions designed for interventions implemented in both educational and clinical settings [23,29,56,57]. Moreover, the available research highlights that the use of AR risks being counterproductive, such as in terms of users' interest caused by novelty overwhelming educational gains [29] or reduced attention with special categories of users, such as children diagnosed with ASD or ADHD [53]. Some papers focusing on interventions delivered in educational settings [22,28] also reported educators' initial hesitancy to integrate AR technology into their consolidated practices. Given these considerations, future studies should assess the users' ability to effectively use these technologies in real-world settings [23,52] and use collected data to design adequate training. Finally, only one paper [50] questioned the cross-cultural generalizability of the developed AR solution and called for future research to give a greater focus on culturally relevant issues.

The second research question of this review allowed deep scrutiny of the impact of the use of AR technologies in users' daily lives. The user, who is engaged in continuous integration of what is digital and real, must adapt to evolving opportunities for interaction and new environmental affordances. As a result, he or she will be pushed to change his or her behaviors and intentions. In this regard, research has offered significant insights into the connection between AR and potential behavioral change. The majority of the existing research studies have focused on the technical and design aspects of AR, but little is known about how the technology affects social interaction. However, research has shown that AR technology can sustain young children's empathetic behavior as it stimulates children's imagination and creativity without causing them to lose touch with reality. It has been observed how the use of AR 3D characters (e.g., a human-like cartoon character) influences the behavioral response of children. In particular, children tend to have and develop a higher sensitivity and empathy to insubstantial agents and imaginary companions. This suggests the use of AR characters that are not visible in reality but visible in augmented reality for stimulating the imagination and creativity of students. It is possible to create AR learning environments where AR 3D characters portray the role of a storyteller, capable of telling stories that involve the transmission of values and positive social behavior. These immersive AR learning environments can help foster empathy between the storyteller and students, which facilitates learning. In this regard, augmented reality would require the creative abilities and skills of developers to accommodate the development of AR objects and content in terms of storytelling. Is our educational system currently addressing the knowledge requirements for this innovative future sector of workforce? In future research, education and professional training should be expressly targeted for this topic.

Moreover, gamification can play a key role in the creation of more engaging playing activities. Examples include AR games that invite students to apply behaviors that remove environmental threats, such as picking up rubbish from the ground and throwing it in the bin. This is accomplished by using play as a lever to promote values and the adoption of positive behavior with a high environmental impact. Furthermore, people who play digital games are more open to improvements in their behavior. Specifically, AR games result in beneficial behavioral effects on users' behavior, including social interaction. Other studies have also investigated the impact of specific modes of presenting augmented information or objects on users' beliefs. For example, augmenting an agent with a visual body in AR and normal social behaviors could enhance the user's confidence in the agent's ability to affect the real world. Furthermore, in education, AR can help with learning difficulties and improve the learning experience. This aspect of AR can be beneficial when working with the requirements of special needs children, who are less likely to spend time and effort on their specific weaknesses.

## 5. Conclusions

The results of this study indicate that the reported suggestions pertain to the potential of AR solutions to support behavioral learning and training. Using AR in education requires programs that integrate both learning skills and teaching skills, which are essential traits. With regard to the use of specific AR tools in the context of behavioral interventions, most of the solutions designed for both educational and clinical purposes are marker-based and use smartphones and tablets. On the contrary, the use of head-mounted devices or more innovative solutions appears, at the moment, to be limited to specific settings or to treating specific disorders. In settings in which a larger number of people is involved, limitations to the adoption of these devices are still relevant. Interesting news may be introduced by software innovations aimed at improving the users' experiences, especially in terms of multi-user interaction. As a result of these enhancements, the user experience may be enhanced not only via immersive features but also through a plurality of interactions and richness of information spread throughout the environment. To conclude, some limitations need to be considered in interpreting the results. First, only papers written in English were selected for inclusion in the present study. As a consequence, we potentially excluded relevant articles published in different languages. Moreover, we searched for papers published from 2010. Even if papers published before this date were automatically excluded from the analysis, we considered that the majority of relevant research on the topics of interest had been published after 2010. Finally, we tried to reduce the risk of publication bias by purposefully searching for the gray literature in OpenAIRE.

**Author Contributions:** Conceptualization, C.T., F.M., L.S., G.C., E.M.; methodology, L.S.; software, L.S.; investigation, F.M., G.C., A.C., M.A., D.T., M.F., E.M.; data curation, F.M., L.S.; writing—original draft preparation, C.T., F.M., L.S., G.C., A.C., M.A., D.T., M.F., E.M.; writing—review and editing, C.T., E.M.; supervision, E.M. All authors have read and agreed to the published version of the manuscript.

**Funding:** This document is the result of the research supported by European Union's Horizon 2020 research and innovation program under grant agreement No. 856533, project ARETE.

**Institutional Review Board Statement:** Not applicable.

**Informed Consent Statement:** Not applicable.

**Data Availability Statement:** Data has been presented in the main text.

**Conflicts of Interest:** The author declares no conflict of interest.

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
