# Peer review of "The Potential of AR Solutions for Behavioral Learning: A Scoping Review"

_computers, doi:10.3390/computers11060087_

Round 1
Reviewer 1 Report
This study investigates the impact of Augmented Reality (AR) in promoting learning in social sciences. Specifically authors performs a scoping review (applying the Prisma methodology) to collect and analyze studies exploiting AR in interventions for teaching behavioural skills to different target populations. The selected papers (N = 86) were grouped by setting of intervention (educational, clinical, professional, others).
The review reports an investigation of literature discussing the impact of AR technology on social behavior. The state of the art of AR solutions designed for interventions in behavioral teaching and learning is presented, with an emphasis on educational and clinical settings. Moreover, some relevant dimensions of the impact of AR on social behavior is discussed in detail.
This topic is very relevant since people are attracted by Augmented Reality (AR) and games, in particular the young population. AR can offer advantages in learning, rehabilitation, behavior modification. This can benefit the whole population but it is crucial in autism syndrome and ADHD.
This study is very interesting since how this review observed the use of AR is increased in last few years, thus the full potential of AR has not been yet discovered. Having a clear vision of the state of art could stimulate progress in this field. This study offers an interesting contribution in the application AR technologies and applications in social science, by highlighting missing research.
The paper is fluent, easy to read and to understand. The number of analyzed paper is adequate for a review. The methodology is solid and well introduced also to reader without previous knowledge. The research questions was adequately addressed and clearly discussed, in a holistic view. Limitations of the reviewed AR solutions and implications for future research and development efforts are finally discussed.
In my opinion this paper is ready to be published and can benefit the reader.
Author Response
Thank you for your consideration of our work, much appreciated. We really hope our paper can be useful to readers with different backgrounds and interests.
Reviewer 2 Report
The paper reviews the applications of augmented reality (AR) technologies in behavioural learning. This topic seems to be interesting and actual. The main ideas are relatively clearly explained. I have only partial comments on the methodology and particularly its description.
I recommend the paper for acceptance after minor revisions. See also the list of comments below.
Methodology:
- Why were different numbers of scientific databases (three versus five) analysed for these research questions?
- Why are the authors not interested in the year and title of the publication in the second research question?
- It would be more appropriate to present the search terms graphically (e.g., by a diagram).
- I would better recommend (for example, in the introduction to the Results section) to specify how the number of papers for RQ1 and RQ2 is related (as well as Fig. 1 and Fig. 2).
Discussion
- The methodology's paragraphs (e.g., 407 – 413) are repeated here; I consider this unnecessary.
Other minor comments:
- Information about the research project is usually given at the end of the paper.
Author Response
The paper reviews the applications of augmented reality (AR) technologies in behavioural learning. This topic seems to be interesting and actual. The main ideas are relatively clearly explained. I have only partial comments on the methodology and particularly its description.
I recommend the paper for acceptance after minor revisions. See also the list of comments below.
Methodology:
Point 1: Why were different numbers of scientific databases (three versus five) analysed for these research questions?
Response 1:
The first research question was specifically dedicated to analysis of intervention designed to teach/train behaviuoral skills. Given this specific focus, we chose to opt for those databases more likely including papers concerning medical, psychological, and educational research on our topic (PubMed, Web of Science, Scopus). We also decided not to use the other two databases (IEEE Xplore, ACM Digital Library), more specialized on technological and engineering issues, to avoid the risk of inefficiently extend the research.
Point 2: Why are the authors not interested in the year and title of the publication in the second research question?
Response 2:
Thank you for your comment. Actually, title and year of publication were extracted and this process has been described for research question 2. On the contrary, title of publication was extracted as well for the first research question but this information is not described in the text of the paper. We apologize; the process is now fully described as extraction of paper title has been specified at page 5, line 132.
Point 3: It would be more appropriate to present the search terms graphically (e.g., by a diagram).
Response 3:
Thank you for the comment. The search terms have been added to Figure 1 and 2 (PRISMA protocols) to further clarify the review process.
Point 4: I would better recommend (for example, in the introduction to the Results section) to specify how the number of papers for RQ1 and RQ2 is related (as well as Fig. 1 and Fig. 2).
Response 4:
We agree with this useful comment, thank you. The list of papers overlapping for RQ1 and RQ2 has been added in the introduction to the Results section at page 5, line 165.
Discussion
Point 5: The methodology's paragraphs (e.g., 407 – 413) are repeated here; I consider this unnecessary.
Response 5:
We appreciated this comment, thank you. Accordingly, we have shortened the introduction of the Discussion section, deleting unnecessary statements.
Other minor comments:
Point 6: Information about the research project is usually given at the end of the paper.
Response 6:
Funding information has been added at page 12, line 516.
Reviewer 3 Report
Great work! I really enjoyed reading this review. The article summarizes a scoping review centered on two topics: 1. how are AR systems designed for teaching a behavior? And 2. the impact of using AR on social behavior. The authors precisely enumerate the protocols for selecting relevant literature, and discuss them from different angles. One thing that could potentially enhance the paper is explaining and discussing more open-end research questions in the Discussion session.Author Response
Point 1: Great work! I really enjoyed reading this review. The article summarizes a scoping review centered on two topics: 1. how are AR systems designed for teaching a behavior? And 2. the impact of using AR on social behavior. The authors precisely enumerate the protocols for selecting relevant literature, and discuss them from different angles. One thing that could potentially enhance the paper is explaining and discussing more open-end research questions in the Discussion session.
Response 1:
We deeply thank you for your appreciation of our work. In consideration of your suggestion, we tried to enrich the Discussion section with the introduction of few additional open-end research questions to further stimulate readers’ attention.
Additionally, we checked the manuscript and did some spell corrections.